# Joint Stiffness Influence on the First-Order Seismic Capacity of Dry-Joint Masonry Structures: Numerical DEM Investigations

**Nathanaël Savalle** [1,*], **Paulo B. Lourenço** [1] **and Gabriele Milani** [2]

1    Institute for Sustainability and Innovation in Structural Engineering (ISISE), Department of Civil Engineering, University of Minho, 4800-058 Guimarães, Portugal; pbl@civil.uminho.pt
2    Department of Architecture, Built Environment and Construction Engineering (ABC), Politecnico di Milano, Piazza Leonardo da Vinci 32, 20133 Milan, Italy; gabriele.milani@polimi.it
*    Correspondence: n.savalle@civil.uminho.pt

**Featured Application: Numerical modelling and seismic capacity of dry-joint masonry structures.**

**Abstract:** Heritage masonry structures are often modelled as dry-jointed structures. On the one hand, it may correspond to the reality where the initial mortar was weak, missing, or has disappeared through time because of erosion and lixiviation. On the other hand, this modelling approach reduces complexity to the studied problem, both from an experimental and theoretical/numerical point of views, while being conservative. Still, for modelling purposes, in addition to the joint friction, numerical approaches require a specific elastic parameter, the dry-joint stiffness, which is often hard to estimate experimentally. This work numerically investigates the effect of the joint stiffness on the collapse of scaled-down tilting test experiments carried out on perforated dry-joint masonry shear walls. It is found that geometrical imperfections of bricks and the absence of vertical precompression load can lead to very low equivalent dry-joint stiffness, which strongly affects the results, both in terms of collapse and damage limit state (DLS) loads, with practical implications for the engineering practice.

**Keywords:** joint stiffness; seismic behaviour; masonry built heritage; tilting tests; Discrete Element Method (DEM)

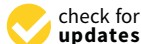



## 1. Introduction

Built masonry structures constitute an invaluable heritage worldwide, including in Europe, where many historical centres and UNESCO heritage sites made with masonry can be found. To this extent, the study of their behaviour is one of the main concerns of the international civil engineering community. Existing masonry buildings are often modelled with an assemblage of masonry units with dry interfaces [1–12]. Firstly, the mortar used at the construction time was often of low quality, and was even missing in some cases. Secondly, the initial mortar may have been degraded through time because of erosion and lixiviation. Finally, this assumption is conservative and enables experimental tests in the laboratories to validate the developed models to be performed more easily [3,4,13–22]. Indeed, dry masonry allows both faster tests (no mortar hardening time) and scaled tests without needing to scale the mortar size while keeping adequate mechanical properties.

Modelling strategies for masonry assemblies depend on the type of structure and the type of loading. While continuum Finite Element Modelling (FEM) strategies have been found useful for modelling large buildings with a relatively good quality mortar [23–25], Discrete Element Modelling (DEM) and similar discrete approaches often perform better for dry masonry structures, allowing large displacement to occur at dry interfaces [2,4,5,8,23]. When referring to the type of loading, one can find settlement [3,4,13,17,21], vertical loading [26] and seismic loading which can then be classified as either pseudo-static (tilting) [6,10,14,20,27,28] or dynamic loading [2,7,9,12].

The seismic vulnerability of masonry structures is particularly high because of their heavy weight, brittleness, low tensile strength and poor connection between structural components [29,30]. As far as the seismic behaviour of masonry structures is concerned, numerical FEM and DEM schemes can be used to perform both types of analysis, pseudo-static (so-called pushover analysis) and dynamic time history [7,12]. On the other hand, analytical approaches have also been derived with the Limit Analysis framework for pseudo-static loading [6,10,31] and rocking theory for dynamic loading [2,9].

Discrete Element Methods adopt an assemblage of highly rigid blocks in contact at interfaces. The deformability of the whole structure is mainly driven by the joints, while the block deformability is usually taken as either null or very limited [7]. In fact, at interfaces, most DEM strategies assume a stiffness that relates the force transmitted through the interface with the elastic displacement at the same interface. The latter is characterised by a relative displacement between the two blocks in contact. Common values used in the literature lead to a global behaviour (i.e., elastic stiffness) driven by the joint stiffness [4,7,12]. In general, the calibration of joint elastic parameters remains complicated, often conducted by inverse fitting, which prevents reliable predictions of structural behaviour. Instead, only post-diction can be carried out. For instance, in dynamic tests, the stiffness parameters can be calibrated through the recorded vibration modes of the structure [7]. Values can also be assumed on a default basis, either taken from the literature or building codes [4,12], but this hardly applies to tests with assembled dry joints blocks and low vertical precompression.

More recently, the joint behaviour of dry masonry assemblages has been more deeply investigated experimentally [32–35]. It has been shown that the joint behaviour was strongly influenced by the contact stress and geometrical imperfections of blocks in contact (non-planarity of faces, height differences between blocks). In particular, joint stiffness strongly diminishes with decreasing vertical stress, with an approximately linear relationship [33,34], while block imperfections reduce the joint stiffness even more [32,35]. In addition, it has been shown that the local joint stiffness (at a single interface) in an assemblage of blocks with a running bond pattern is much smaller than the joint stiffness in a classical joint closure test with two-stacked blocks [35]. The authors attributed this difference to block heterogeneities (e.g., misalignment and height differences of blocks).

Currently, DEM simulations tend to focus on perfect blocks with relatively high joint stiffnesses, which may not reproduce the behaviour of low-stress assemblages of highly imperfect masonry units. Therefore, the paper aims at modelling, with the Discrete Element Method (DEM), a dry assemblage of masonry units that fits the mentioned conditions. The case study, described in detail in [22], is a masonry shear wall which encompasses twenty-seven courses of bricks, increasing the structure's global flexibility because of the numerous block interfaces. Furthermore, insights about the required stiffness values to be used are given. Section 2 briefly describes the experimental campaign. Section 3 details the chosen DEM methodology. The numerical results and the sensitivity analysis to modelling parameters are presented in Section 4. Section 5 discusses the implications of the numerical results for the engineering practice with insights into the damage limit state (DLS) defined through the drift limits of the Eurocode 8 [36]. Finally, Section 6 gathers the main conclusions and implications of the work.

## 2. Tilting Tests on Perforated Dry-Masonry Walls

This section briefly describes the experimental campaign: for a detailed description, the reader is referred to the two companion papers [22,27], which review the experimental study and detail the numerical approach used to model them, respectively. The campaign consists of seven small-scale walls, later referred to as wall design (WD), with different block arrangements (Figure 1) tested on a tilting table and proposed by different competing teams. The walls were built using the same bricks (length × height × width = 25 mm × 5.5 mm × 12 mm) of density $\rho = 1800 \, \text{kg/m}^3$ and had identical dimensions (length = 0.2 m and height = 0.148 m). Finally, their design had to fill two constraints (Figure 1):

- Having unperforated external edges (two sides and top).
- Having at least 30% of voids inside the structure.

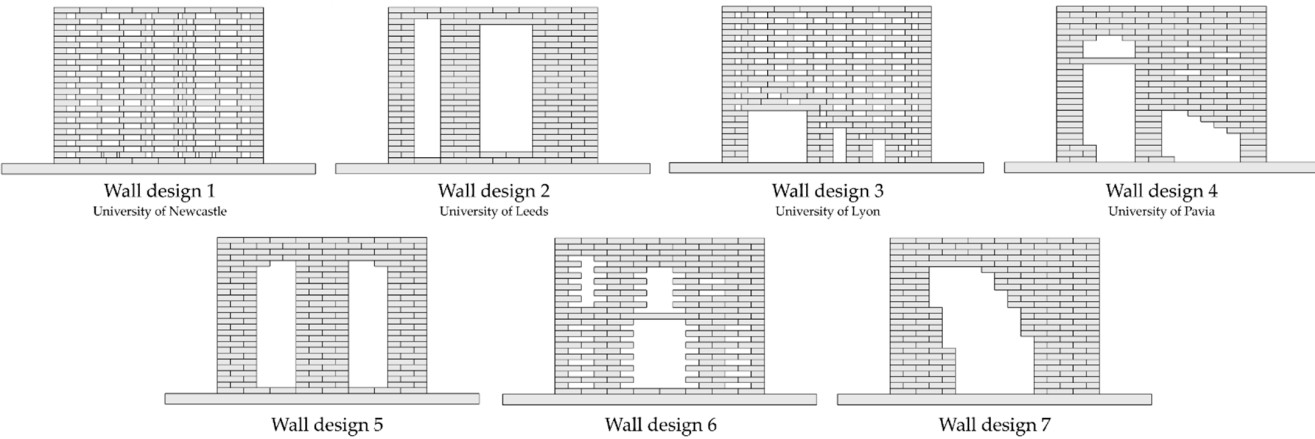

**Figure 1.** Seven wall geometries tested on a tilting table and respective teams, adapted from [22]. The external boundaries (length and height) of each wall design (WD) are the same.

With this in mind, the objective of the design was to reach the maximum resistance to tilting tests, leading to a void ratio very close to 30% for all designs (max 31.5% for wall designs WD 2 and WD 5). Table 1 summarises the collapse tilting angles of each configuration. Tests have been repeated three times, and the coefficient of variation is also indicated in the results. The friction coefficient has been evaluated to $\mu = 0.5$ in the original study. The authors also noticed that the blocks were not geometrically perfect [22], which led to some refinements in the numerical paper [27].

**Table 1.** Experimental collapse tilting angle for each wall design (WD) [22]. The coefficient of variation is also depicted.

|  | WD 1 | WD 2 | WD 3 | WD 4 | WD 5 | WD 6 | WD 7 |
|---|---|---|---|---|---|---|---|
| Collapse tilting angle $\theta$ (°) | 18.3 | 7.8 | 19.8 | 16.8 | 7.8 | 7.8 | 18.6 |
| Coefficient of Variation (%) | 4.7 | 3.2 | 6.2 | 8.1 | 16.8 | 6.0 | 7.1 |

A numerical limit analysis program (with associative and non-associative flow rules) was used to model the experiments [27]. Firstly, assuming blocks as perfectly rectangular, the model largely overestimated the experimentally obtained capacities. The authors then conducted a parametric analysis on the friction coefficient, installation defects (misalignment of bricks) and geometric imperfections of bricks (corner rounding). Indeed, concerning the last point, They noticed that the real contact area was much smaller than the entire block surface, with an averaged reduction of 60%. While the first two had a negligible influence on the numerical results for most tests, the last one had a significant effect. However, from the results, different levels of geometrical imperfections (characterised by different reduction in the contact areas) for each wall design (WD) were necessary to approximately fit the experimental results, while the bricks were the same for the experimental campaign. Therefore, the following sections of the paper focus on discrete element simulations of the same experimental campaign. Therein, the joint stiffness parameters are assumed to account for the blocks' geometrical imperfections.

## 3. Numerical Discrete Element Method (DEM)

The numerical simulations have been conducted on the Discrete Element environment offered by 3DEC [37]. This section summarises the general features of 3DEC related to the specificities of the present numerical simulations.

The considered specimens are described using the so-called simplified micro-modelling strategy. Each block is modelled in 3D by a separate entity, while zero-thickness interfaces model the dry joints. In 3DEC, blocks can be either deformable or rigid. In the present paper, given the low stresses acting on them, blocks are considered fully rigid and characterised by only six degrees of freedom and a unique mechanical parameter: their density ($\rho$ = 1800 kg/m$^3$). Furthermore, a single numerical simulation using deformable blocks with very low elastic rigidity (E = 1 $\times$ 10$^8$ Pa) confirmed the negligible influence of the block deformation itself.

At the interface between blocks (either for bed or head joints), contact forces are modelled through equivalent springs (Figure 2). One normal spring relates the elastic block interpenetration in the normal direction of the joint ($u_n$) with the normal stress ($\sigma$). Similarly, two tangential springs relate the shear elastic relative displacement ($u_s$) with the shear stresses ($\tau$) in the two tangential directions. Equation (1) describes the link between the increments in stress and displacement, in the process of moving from step t to step t + $\Delta$t. Only one shear equation is shown for brevity, instead of the two implemented in 3DEC.

$$\Delta\sigma = k_n \cdot \Delta u_n = k_n \cdot [u_n(t + \Delta t) - u_n(t)]$$
$$\Delta\tau = k_s \cdot \Delta u_s = ks \cdot [u_s(t + \Delta t) - u_s(t)]. \tag{1}$$

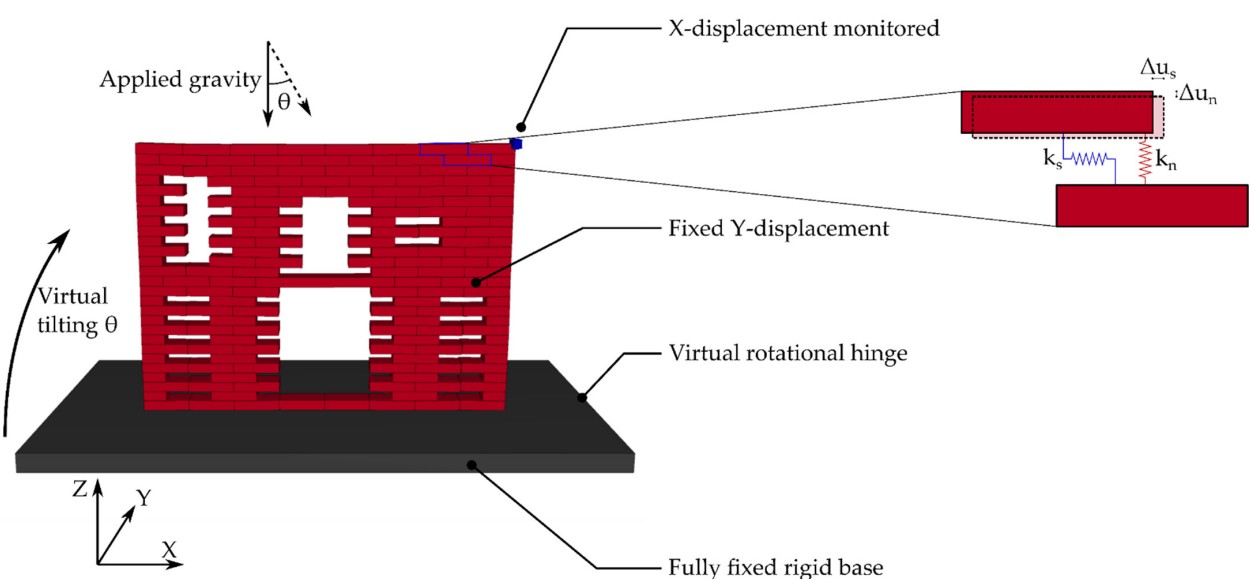

**Figure 2.** WD 6 after vertical gravity stabilisation ($k_n = k_s = 3 \times 10^7$). Boundary conditions and reference axis are shown. The monitored point is also shown (see Section 5).

Here, $k_n$ and $k_s$ are the joint's normal and tangential elastic stiffness and are sufficiently large to avoid unrealistic interpenetration of blocks. The experimental characterisation of joint stiffness parameters is not provided in [22]. Therefore, their values are varied through a parametric analysis presented in the next section. When two blocks are automatically determined to be in contact, a few contact points are generated, and the contact forces are transmitted discretely at these locations. By default, 3DEC creates contact points at each vertex of blocks, but it has been found that more contact points were required to model the actual stress distribution accurately [7]. This issue is further discussed in the next section concerning the modelled experiments.

The Discrete Element algorithm uses an explicit algorithm to solve the equations of motion. At a given step t, all forces (including contact, body and other applied forces) and position of blocks are known. For each block, the equations of motion are integrated to compute the new acceleration of blocks (Equation (2)). Afterwards, a central difference

algorithm calculates the block velocities, which enable finding the new block positions at step t + $\Delta$t.

$$m_i a_i(t) = \Sigma F_i(t)$$
$$v_i(t + \Delta t/2) = v_i(t - \Delta t/2) + a_i(t) \cdot \Delta t \qquad (2)$$
$$u_i(t + \Delta t) = u_i(t) + v_i(t + \Delta t/2) \cdot \Delta t.$$

Here, $u_i$, $v_i$, $a_i$, $m_i$ and $F_i$ are the position, velocity, acceleration, mass and applied forces of block i. New block positions generate increment in the contact forces (Equation (1)). At this stage, the constitutive law of the joint is applied to update the contact force. In the case of the studied masonry specimens, a classical Mohr–Coulomb behaviour with the experimental friction coefficient $\mu$, no tension, no cohesion, and no dilatancy has been considered [38]. It reads:

$$\sigma(t + \Delta t) = \sigma(t) + \Delta\sigma, \sigma < 0 \text{ (only compression)}$$
$$\tau(t + \Delta t) = \tau(t) + \Delta\tau, |\tau| < \tau_{max} = -\sigma \cdot \mu. \qquad (3)$$

Note that the convention used in Equation (3) leads to compression stresses displayed as negative numbers. Stresses are then multiplied by the associated contact areas, and new contact forces are derived and are inputted into Equation (2) for the next timestep. Such an integration scheme requires that the timestep $\Delta$t is chosen to be sufficiently small, related to the maximum ratio between local stiffness and local mass, and is automatically computed in 3DEC. In addition, DEM simulations are often artificially damped for static analysis to avoid oscillations around the equilibrium position and to speed-up the analysis. This work considers the classical local damping of $\xi$ = 0.8 [4,37], which reduces the acceleration $a_i$, replacing the first equality shown in Equation (2) by:

$$m_i a_i(t) = \Sigma F_i(t) - \xi \cdot |\Sigma F_i(t)| \cdot \text{sign}(v_i(t - \Delta t/2)). \qquad (4)$$

From Equation (4), one can note that the damping force is always opposed to the velocity of the computational node.

For each wall design (WD), the model has been directly imported from the Rhinoceros3D environment through the VRML format ("block generate from-vrml" command) [37,39]. Then, an additional block, called "base", is added to support the whole model: its six degrees of freedom are fixed (Figure 2). The contact interface between the base block and the first course of the masonry model is similar to the interface between blocks, as no information is provided in [22] to support other modelling strategies. In addition, no sliding of the first course of blocks has been noticed during both the experiments [22] and the simulations (Section 4), validating the accuracy of the modelling strategy. All blocks were also fixed in the Y-direction as no movement was allowed in this direction during the experiments [22]. Finally, all the material properties (density and interface parameters) are applied to the model.

As a first stage, static loading with only vertical gravity is activated, and the model runs until stability is achieved. This is assumed when the maximum velocity in the model is below a threshold of $1 \times 10^{-13}$ m/s. Applying vertical gravity with 100 progressive steps or a single one was not affecting the results (i.e., the collapse tilting angle $\theta$), given the low nonlinear effects in gravity loading for the different designs. Therefore, the most time-efficient strategy (single-step) was used.

In a second stage, gravity is progressively inclined by steps of 0.1°, according to the formula:

$$\theta = \theta + 0.1°$$
$$\vec{g} = g \times \left( \sin(\theta)\vec{x} - \cos(\theta)\vec{z} \right). \qquad (5)$$

Here, g = 9.81 m/s$^2$ is the intensity of gravity and the inclination of gravity $\theta$ corresponds to the inclination of the tilting plane in the experiments. For each tilting increment, the system waits until stabilisation before moving to the next step, using the threshold defined in the first stage. The simulation runs until the horizontal displacement of the top

block reaches a limit (Figure 2), here chosen as 10 cm (i.e., half the model length). One can note that using non-constant increments of the inclination angle (similarly as in [12]) could reduce the computational time of the simulations. The files used for the simulation and the 3D models are made publicly available [40].

## 4. Results

### 4.1. Mesh Sensitivity Analysis

The stress distribution in DEM is better reproduced when using more contact points [7]. Then, a balance needs to be found between accuracy and requested time for practical applications. A preliminary analysis has been conducted using the wall design WD 3 and Figure 3 shows the influence of the number of contact points along full block length on the collapse tilting angle θ. Eight contact points provide a difference of 1% with respect to the previous discretisation and may be considered an exact solution. The four contact points models have been considered accurate enough (difference less than 4%) for the parametric analysis of the stiffness parameters (Sections 4.2, 4.3 and 5), especially given the high requested time to run the simulations. Further simulations using other WDs (not shown here for brevity) confirm the minor difference between four and more contact points.

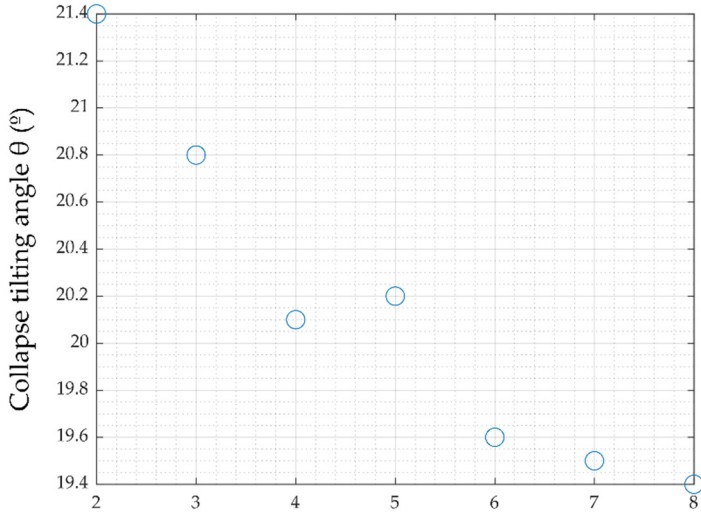

**Figure 3.** Effect of the density of contact points used in the numerical DEM simulations on the collapse tilting angle θ.

### 4.2. Simulations with Classical Joint Stiffness Values

First, simulations have been carried out using classical values for dry-joint stiffness. In the literature, joint stiffness is often calibrated to account for the deformability of masonry units and mortar [4,12,41], even when modelling mortarless masonry structures [4] to account for geometric irregularities. In other cases, the joint stiffness is based only on the deformability of blocks [42,43]. Finally, some studies take arbitrary, though reasonable, values [44,45]. According to the references listed above, acceptable values for dry-joint behaviour range from $5 \times 10^8$ to $1 \times 10^{11}$ Pa/m. The adopted ratio between $k_s$ and $k_n$ varies between 0.4 and 1.0. Initially, two simulations were run for each model with the following parameters:

- $k_n = k_s = 1 \times 10^9$ Pa/m
- $k_n = k_s = 1 \times 10^{10}$ Pa/m.

The ratio $k_s/k_n$ is taken as equal to 1.0, though different ratios do not seem to influence the results significantly (see Section 4.3). The collapse tilting angles θ obtained for each simulation are gathered in Table 2. The original results of the Limit Analysis model are also shown [27], including associative and non-associative solutions. One can observe that the

results provided by the different approaches are rather close (larger deviation in case of WD 1), even if quite different from the experimental results.

**Table 2.** Collapse tilting angle θ (°) obtained for different modelling strategies. The results of the Limit Analysis with associative (LA-A) and non-associative (LA-NA) flow rules [27] and DEM simulations for two different joint stiffness $k_n = k_s = 1 \times 10^9$ and $1 \times 10^{10}$ Pa/m are presented. The experimental tilting capacities are also indicated for reference.

| | Exp [22] | LA-A [27] | LA-NA [27] | DEM ($1 \times 10^9$) | DEM ($1 \times 10^{10}$) |
|---|---|---|---|---|---|
| WD 1 (Newcastle) | 18.3° | 23.9° | 23.5° | 25.3° | 26.5° |
| WD 2 (Leeds) | 7.8° | 17.0° | 16.0° | 15.5° | 16.9° |
| WD 3 (Lyon) | 19.8° | 26.5° | 24.5° | 25.6° | 26.3° |
| WD 4 (Pavia) | 16.8° | 24.6° | 23.5° | 23.8° | 24.9° |
| WD 5 (Leuven) | 7.8° | 17.5° | 17.4° | 16.4° | 18.0° |
| WD 6 (Yildiz) | 7.8° | 18.3° | 17.1° | 15.9° | 17.4° |
| WD 7 (Munich) | 18.6° | 26.6° | 26.6° | 26.5° | 27.0° |

The reason for the differences found is that limit analysis solutions do not consider joint (or block) deformation under applied stresses and progressive changes in the structure due to load incrementation. For example, limit analysis fails to reproduce the typical progressive bending behaviour observed for slender masonry piers (Figure 4).

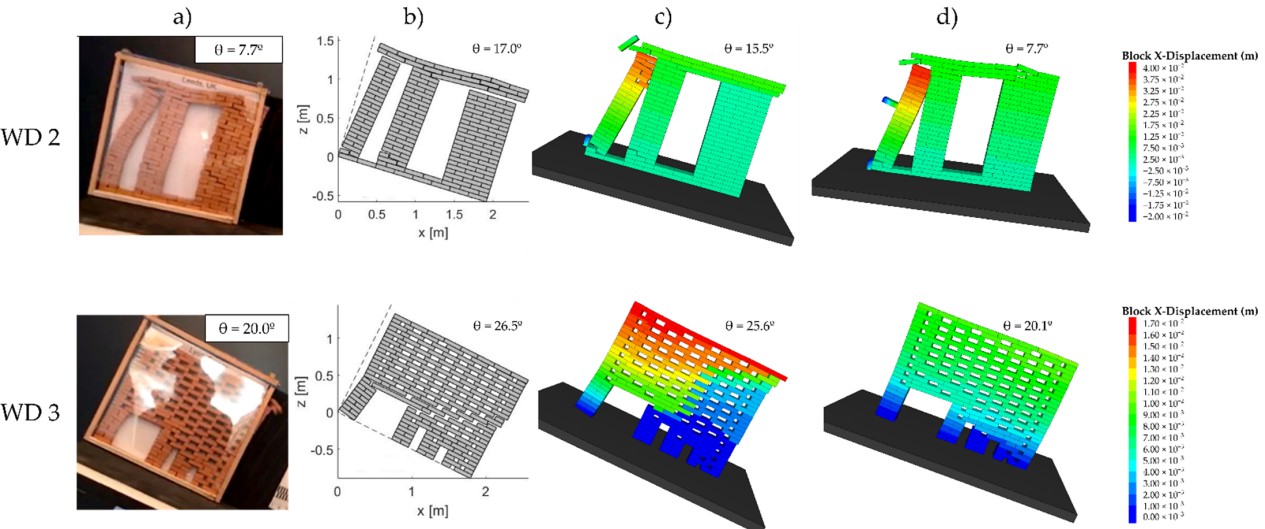

**Figure 4.** Collapse mechanisms obtained for different modelling strategies on two WDs: (**a**) Experimental failure [22]; (**b**) Limit Analysis with associative behaviour [27]; (**c**) DEM simulations with $k_n = k_s = 1 \times 10^9$ Pa/m; (**d**) DEM simulations with $k_n = k_s = 3 \times 10^7$ Pa/m. The reader is specially referred to the left masonry pier bending.

The predicted failures of the columns are too rigid when compared to the experimental ones. Similarly, 3DEC simulations lead to the same type of rigid failure, indicating that the chosen numerical parameters are not appropriate. As already noted by others [42,43], further increasing the joint stiffness does not significantly increase the tilting capacity (Table 2). On average, these "rigid" simulations overestimate the tilting capacity by a factor of 1.7 (considering the experimental values as references), which drops to a factor of 1.4 for the more performant WDs (WD 1, WD 3, WD 4, WD 7) but rises to 2.2 when looking at the less performant ones (WD 2, WD 5, WD 6). Instead, as illustrated in Figure 4d, using lower joint stiffness values can provide much better results (Section 4.3).

As a final remark, WD 1 shows more discrepancies between the Limit Analysis results and 3DEC simulations (Table 2). This is due to the local collapse noted in Limit Analysis (Figure 5), which happens without any global collapse of the wall. Similarly, 3DEC sim-

ulations also predict this local failure at approximately the same tilting angle θ but then continue up to the more extensive collapse of the structure.

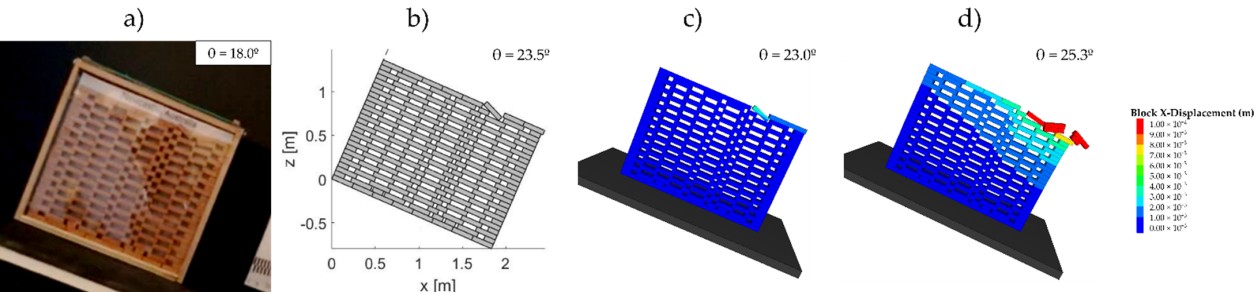

**Figure 5.** Collapse mechanism observed for WD 1. (**a**) Experimental [22]; (**b**) Limit Analysis (Non-Associative) [27]; (**c**,**d**) DEM simulations with $k_n = k_s = 1 \times 10^9$ Pa/m for different tilting angles θ. Note that both modelling strategies predict the failure of one brick of the last row, but, contrary to LA, DEM can further increase the tilting angle to get the global collapse.

### 4.3. Influence of Low Joint Stiffness on the Simulation Results

To decrease the global stiffness of the model, the interface stiffness is now reduced (Figure 4). As noted in Section 3, given the low stress acting in the experiments, block deformation is unlikely to affect the results significantly: therefore, blocks were kept rigid. On the contrary, it is known that low stresses lead to lower joint stiffness [33,34,46]. As an example, the formulas found by [33,46] for other materials would give the following stiffness at the low vertical stress of the experiments ($\sigma$ = 2.6 kPa at the bottom of the model):

- $k_n = 17.52 \times 2.6 = 4.6 \times 10^7$ Pa/m [46]
- $k_n = 8.8–30.4 \times 2.6 = 2.3 \times 10^7–7.9 \times 10^7$ Pa/m [33]

Obviously, these values have been obtained for other materials, and one could argue that the fitting carried out in [33,46] does not perfectly match the initial curve of the joint closure tests. However, at least it gives reference values for numerical purposes. In addition, block imperfections (non-planarity and large asperities) have been acknowledged to further decrease joint stiffness [32,35]. As the contact area decreases, the local stress increases locally, thus the deformation increases [32]. More importantly, [35] brought to light that the actual local normal joint stiffness in a masonry wall is less than that identified through classical joint closure tests because of the misalignment of staggered blocks (height differences) and geometrical imperfections of the blocks themselves (non-parallelism of faces). This last comment totally applies to the considered experimental campaign [22]. From the results in Oliveira et al. [35], the ratio between normal stiffness in the wall and normal stiffness of joint closure tests lies between 2 and 3.

Finally, a few numerical studies focused on dry-stone masonry walls assembled with rubble stones. Given the highly imperfect geometries of stones, they used relatively low stiffness values ($k_n = 5 \times 10^7–1.25 \times 10^8$ Pa/m) to reproduce field experiments [47–49]. Imperfections in block geometry and block contacts seem to be easily linked with interface stiffness parameters.

Based on this bibliographic information, a parametric study of joint stiffness parameters from $k_n = 1 \times 10^7$ Pa/m until $k_n = 1 \times 10^{10}$ Pa/m is carried out. From several simulations at $k_n = 1 \times 10^7$, $4 \times 10^7$, $7 \times 10^7$, and $1 \times 10^8$ Pa/m, the tangential stiffness has shown very little effect on the collapse tilting angle θ, except small changes (max 1.5°) when increasing $k_s$ from $1 \times 10^7$ to $2 \times 10^7$ Pa/m. For instance, when increasing the tangential stiffness from $4 \times 10^7$ Pa/m to $1 \times 10^8$ Pa/m, the collapse tilting angle is only slightly increased by maximum 0.3° for each WD, considering different levels of normal stiffness ($1 \times 10^7$, $4 \times 10^7$, $7 \times 10^7$, $1 \times 10^8$ Pa/m). For this reason, tangential stiffness has always been considered equal to the normal stiffness $k_s = k_n$, to simplify the parametric analysis, though the tangential stiffness may have a larger effect on other similar structures.

Each wall design (WD) has been numerically simulated using the following stiffness parameters: $k_n = k_s = 1 \times 10^7, 2 \times 10^7, 2.2 \times 10^7, 2.5 \times 10^7, 2.7 \times 10^7, 3 \times 10^7, 4 \times 10^7, 7 \times 10^7, 1 \times 10^8, 1 \times 10^9$ and $1 \times 10^{10}$ Pa/m. For simplicity, the normal stiffness has been kept constant throughout the simulations, though in theory it should be decreased while the model is tilted since the axial stresses decrease. An even better improvement consists of implementing in the numerical model a stress-dependent joint stiffness, but this is out of the scope of the present paper. The collapse tilting angles are represented in Figure 6, in a dimensionless form, dividing the numerical value by the experimental one. In Figure 6, the mean experimental coefficient of variation (CoV) is also represented with the grey shadow for a better appreciation of the quality of the numerical results. On the left side of Figure 6, it seems that joint stiffness higher than $k_n = 1 \times 10^{10}$ Pa/m would not lead to significant changes in the collapse tilting angle $\theta$. Four WDs (WD 1, WD 3, WD 4 and WD 7) saturate even before (at $k_n = 1 \times 10^9$ Pa/m) because the failure is already a sliding failure at this stage.

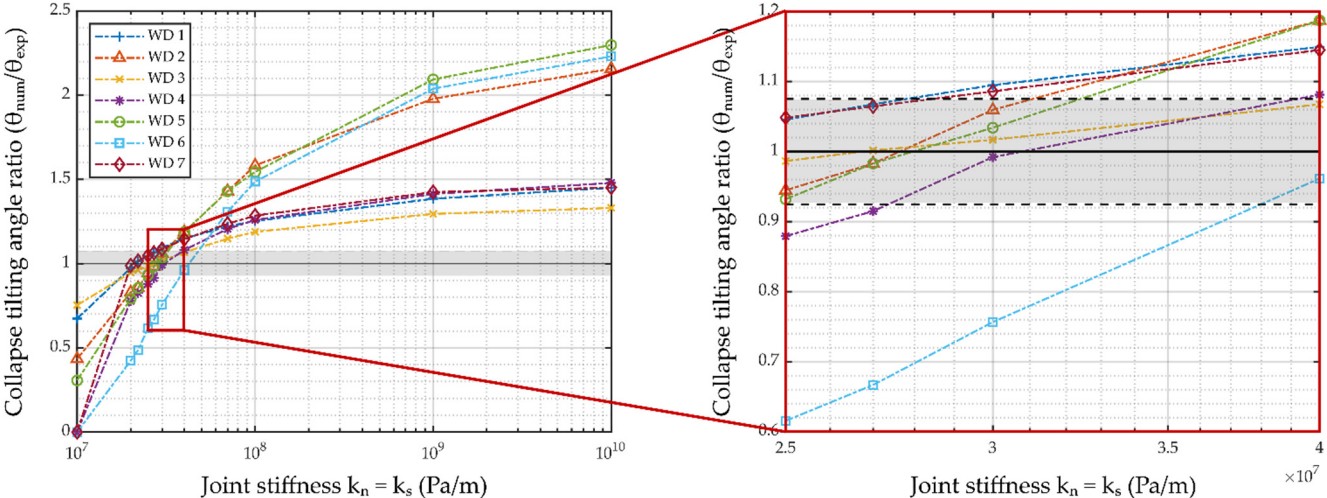

**Figure 6.** Collapse tilting angle $\theta$ depending on the joint stiffness parameters ($k_n = k_s$). The trend is monotonous and indicates a strong effect of joint stiffness.

On the left side of Figure 6, one can note that at $k_n = 1 \times 10^7$ Pa/m, some WDs (WD 4, WD 6 and WD 7) are not even stable under static vertical loads only. Between these two extremes, collapse tilting angles increase with joint stiffness. In general, one can note that numerical results are very close to the experimental ones when the stiffness ranges between $2.5 \times 10^7$ Pa/m and $4 \times 10^7$ Pa/m for each WD.

From Figure 6, the value $k_n = 3 \times 10^7$ Pa/m is excellent for predicting the experimental behaviour accurately. In particular, its accuracy appears to be much better than the one obtained with limit analysis, where different block geometries should be used for each WD to fit the experimental results [27]. Here, the same geometrical, physical and mechanical parameters are used for each WD, leading to an average error of 7.9%: Table 3 sums up the relative difference between numerical and experimental results for this particular stiffness.

The obtained collapse mechanisms for each WD have been compared to the experimental ones (three tests for each WD) in Figures 7 and 8, while the videos of the entire numerical simulations are available in the Supplementary Materials. DEM simulations reproduce the mechanisms well and, in particular, the progressive bending of masonry piers, which is more accurately modelled than using DEM with large joint stiffness values (Section 4.2) or with Limit Analysis [27] (Figure 4). However, apart from these progressive bending effects and the large differences in capacity found as a function of the stiffness, the failure mechanisms are not significantly affected by the stiffness parameters, as already noticed by other authors for dry-joint assemblies [42].

**Table 3.** Comparison between experimental and numerical results with $k_n = k_s = 3 \times 10^7$ Pa/m. The experimental coefficient of variation is also shown for reference. The "Abs average" corresponds to the average of the absolute values.

|  | $\alpha_{exp}$ (°) | $CoV_{exp}$ (%) | $\alpha_{num}$ (°) | Rel. Error (%) |
|---|---|---|---|---|
| WD 1 (Newcastle) | 18.3° | 4.7% | 20.0° | 9.5% |
| WD 2 (Leeds) | 7.8° | 3.2% | 8.3° | 6.0% |
| WD 3 (Lyon) | 19.8° | 6.2% | 20.1° | 1.7% |
| WD 4 (Pavia) | 16.8° | 8.1% | 16.7° | −0.8% |
| WD 5 (Leuven) | 7.8° | 16.8% | 8.1° | 3.8% |
| WD 6 (Yildiz) | 7.8° | 6.0% | 5.9° | −24.7% |
| WD 7 (Munich) | 18.6° | 7.1% | 20.2° | 8.6% |
| Abs average (%) | - | 7.5% | - | 7.9% |

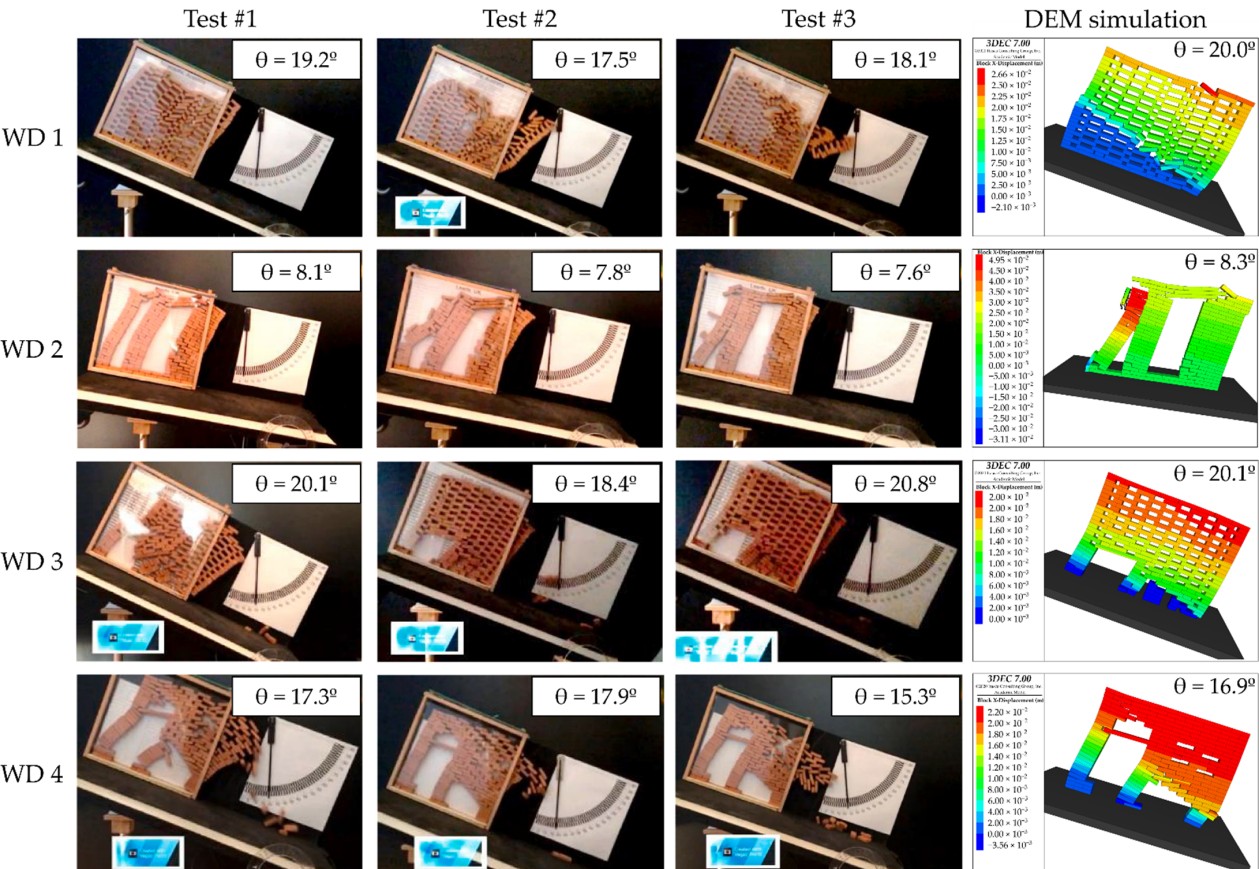

**Figure 7.** Comparison of the observed collapse mechanisms between experiments and DEM simulations for $k_n = k_s = 3 \times 10^7$ Pa/m for WD 1 until WD 4 [22].

Given the strong influence of $k_n$ on the collapse tilting angle θ (Figure 6), the less accurate simulation prediction presented for the model WD 6 is not significant (Figure 8). In addition, one could argue that the stiffness in this particular case might be higher than for the other WDs. Indeed, in the bottom part of the wall where most of the deformation occurs (Figure 8), the contacts are half of the time only between two bricks. Therefore, contact stiffness might be slightly higher than the other cases [35], where contacts mostly consist of one brick lying on two (or more) different bricks (Figure 1).

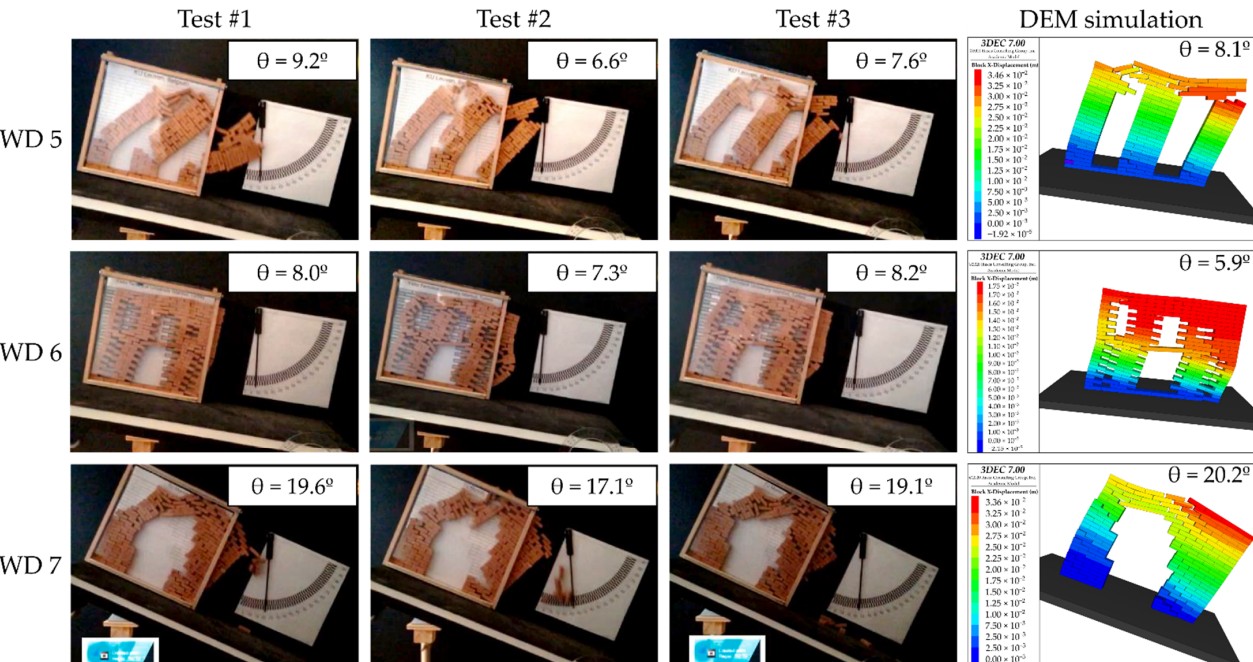

**Figure 8.** Comparison of the observed collapse mechanisms between experiments and DEM simulations for $k_n = k_s = 3 \times 10^7$ Pa/m for WD 5 until WD 7 [22].

As a final remark, it is noted that the four WDs that performed well during the experimental campaign follow the same flat trend in Figure 6 (between $k_n = 3 \times 10^7$ and $k_n = 1 \times 10^{10}$ Pa/m), while the three others also follow another more inclined identical trend. It means that an appropriately chosen design can reduce the effect of joint deformation on the tilting capacity, WD 3 performing the best to this aim with a relative increase of only 80% between $k_n = 1 \times 10^7$ Pa/m and $k_n = 1 \times 10^{10}$ Pa/m.

### 4.4. Validation of the Approach

To summarise the findings shown above, the developed numerical model associated with a joint stiffness of $k_n = k_s = 3 \times 10^7$ Pa/m accurately models the experimental results, both in terms of collapse load multiplier and failure mechanisms. It is important to note that this agreement has been noticed for very different assemblages (WD), with a different distribution of voids (Figure 1), enhancing the consistency of the modelling. In particular, the block imperfections (imperfect contact area) noted in the experimental study [22] can indeed be related to a softer joint stiffness, possibly even more marked because of the low stresses acting throughout the specimens.

When facing such low contact stiffness, it has been noted that the contact stiffness significantly influences numerical results. Therefore, practical steps to assess these soft structures are described hereafter:

- First, the joint stiffness of the studied block contacts must be evaluated. This can be done either using the proposed approach, i.e., calibrating the numerical value based on experiments conducted on structures (post-diction). Ideally, different structures should be investigated. Another option, which is even better, consists in characterising the joint stiffness itself, monitoring joints from a wall under vertical compression, as in [35]. The latter can be complemented by simpler joint closure tests [33]. Note that a combination of both experimental characterisation and numerical calibration/validation is ideal.
- Then, the numerical model with the calibrated parameters can be used in the engineering practice to assess every structure made of the same blocks (predictions).

## 5. Code Aspects and Masonry Structures with Soft Joints

This section compares the pushover curves obtained through the simulations described in the previous section. It also provides insights about the required drift limits for masonry shear walls according to the Eurocode 8 [36].

The previous section evidenced that the experimental collapse multiplier was better reproduced using soft joints rather than more rigid joints. This section first qualitatively extends this comparison to the global stiffness of the structures. Figures 9–11 show the pushover curves for each simulation presented above. Out of all the results, the curves corresponding to a joint stiffness of $1 \times 10^9$ and $1 \times 10^{10}$ Pa/m seem too rigid (less than 1 mm of horizontal displacement at failure) compared to the pictures shown in the reference experimental work [22], see Figures 7 and 8. Based on the videos of the tests presented in [22], the ultimate stable displacements of the walls have been qualitatively estimated to equal 10 mm, without any noticeable variations for all WDs, given the precision of the measurement. Though a stronger comparison would need the experimental pushover curves, which are lacking, this further indicates that the joint stiffness should be lower. Again, a joint stiffness of around $3 \times 10^7$ Pa/m seems an excellent candidate for reproducing these ultimate stable displacements.

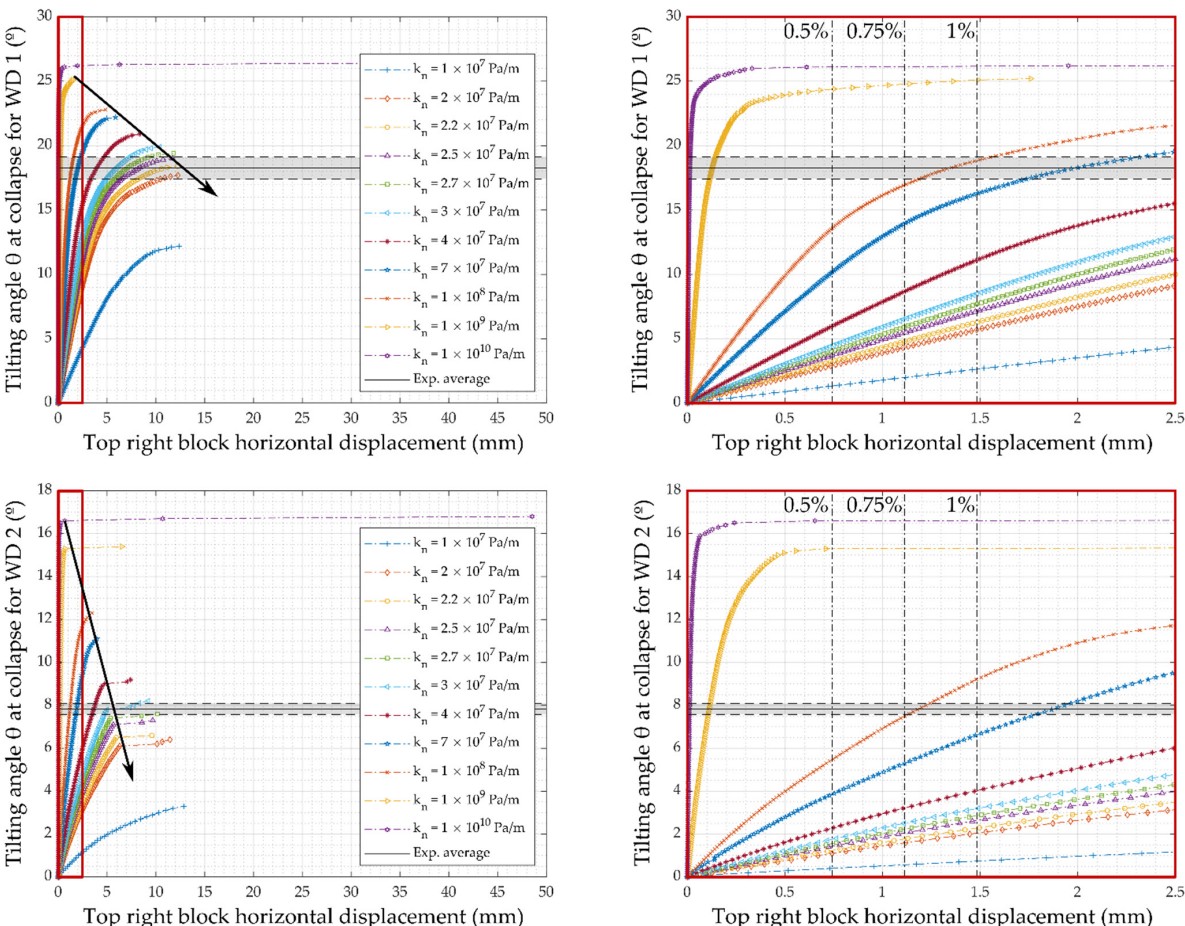

**Figure 9.** Pushover curves obtained for WD 1 and WD 2. Each curve represents a different joint stiffness.

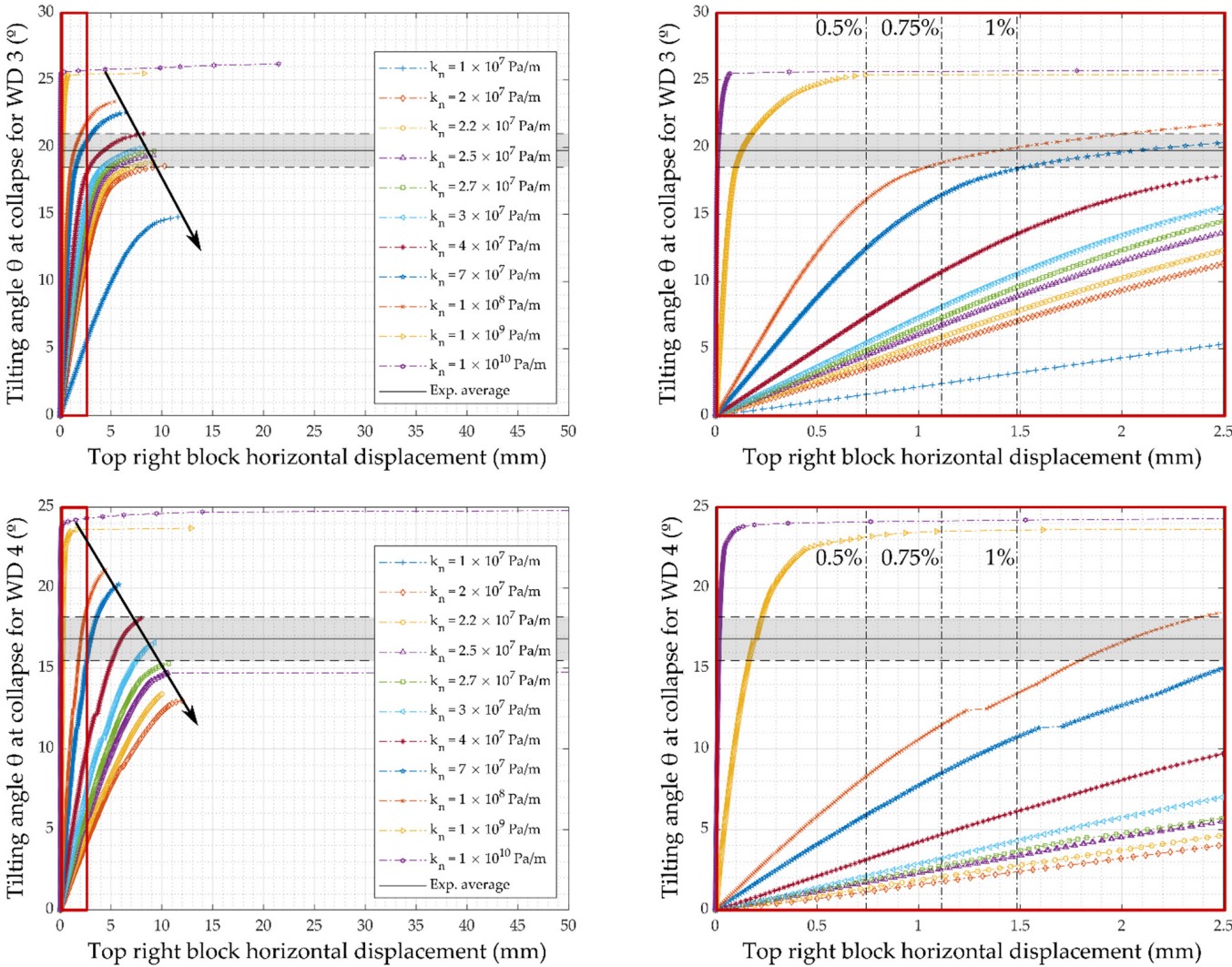

**Figure 10.** Pushover curves obtained for WD 3 and WD 4. Each curve represents a different joint stiffness.

In general, and as expected, the joint stiffness completely drives the stiffness of the whole structure leading to elastic (or at least stable) significant drifts. In fact, from the entire pushover (PO) curves (Figures 9–11 on the left), one can notice that the ultimate stable displacement follows a clear linear trend (indicated by the arrows) for each different WD. For smaller joint stiffness, the deformation is higher and thus leads to larger rotations of the shear walls (though remaining mainly elastic). This geometric change makes the centre of mass of the rocking part of the shear wall closer to the toe and therefore reduces the collapse tilting angle, as a classical rocking problem would do. Though the mechanisms happening in the present shear walls might be more complex than a classical rocking problem, the obtained outcomes are very close to the well-known capacity curves of rocking specimens [50,51]. For each WD, the trend is however very different, mainly due to the various geometrical configurations of the rocking part, indicating that optimisation can be conducted to obtain the "best" capacity curve. In addition, the larger displacements between steps observed at the end of the simulations on Figures 9–11 are attributed to either (1) sliding between blocks, especially for high tilting angles (see WD 1, WD 3, WD 4 and WD 7) or (2) modification of the monitored rocking pier due to the partial collapse of another pier (see WD 2 and Figure 7).

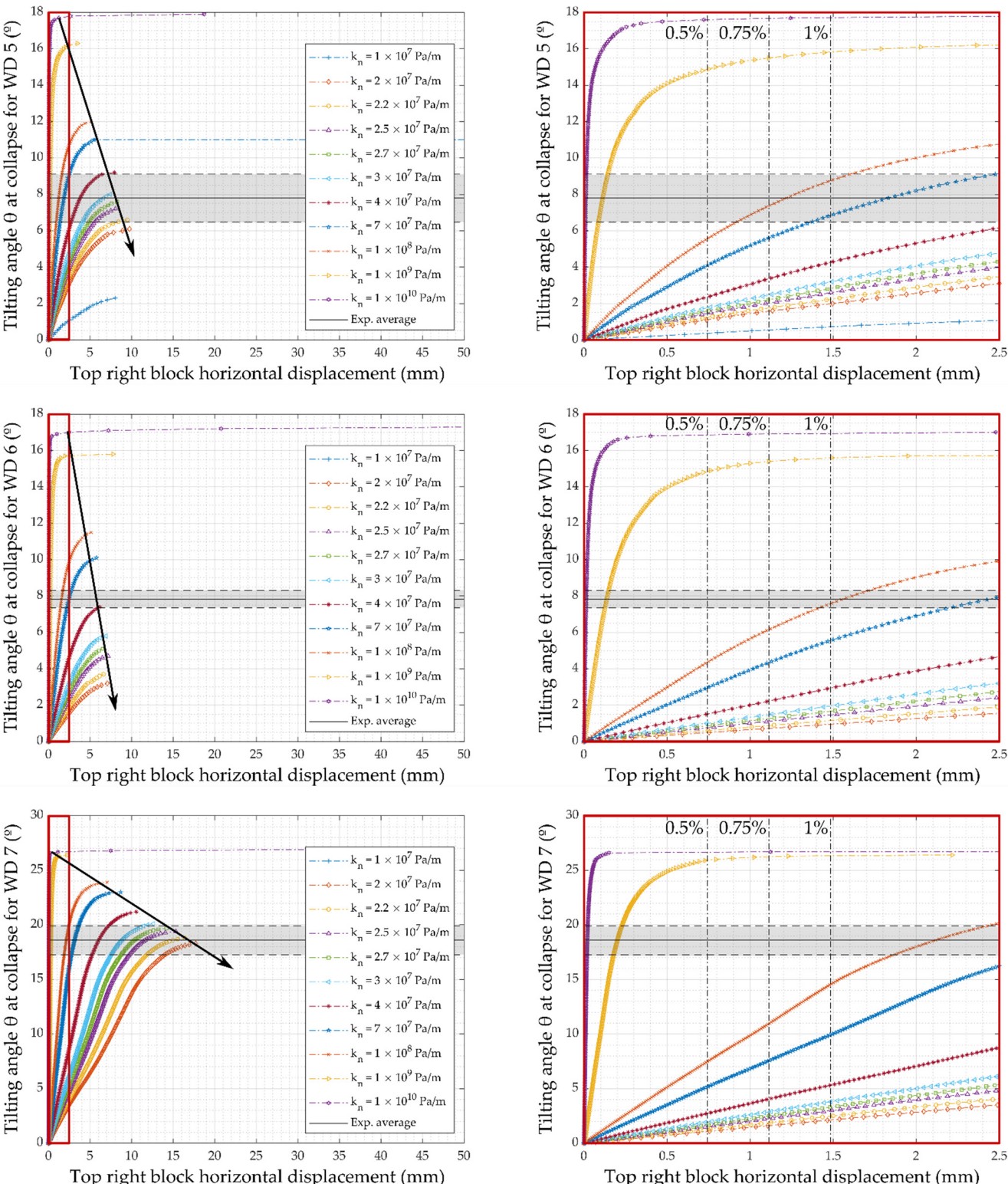

**Figure 11.** Pushover curves obtained for WD 5, WD 6 and WD 7. Each curve represents a different joint stiffness.

In Figures 9–11, the classical drift limits (0.5%, 0.75%, 1%) proposed by the Eurocode 8 [36] are also depicted for all the WDs. Remember that they all hold the same height. For most of the joint stiffness used, except for $k_n = 1 \times 10^9$ and $k_n = 1 \times 10^{10}$ Pa/m, these displacement limits happen in the elastic part of the pushover (PO) curves. For each WD,

Figure 12 shows the evolution of the Damage Limit State (DLS) tilting angle with respect to the contact stiffness. The DLS tilting angle is defined as the tilting angle leading to a drift of 1%, which corresponds to classical structural shear walls without any bound non-structural elements. According to the Eurocode 8, the drift should never exceed this limit when subjected to large probability of occurrence earthquakes [36].

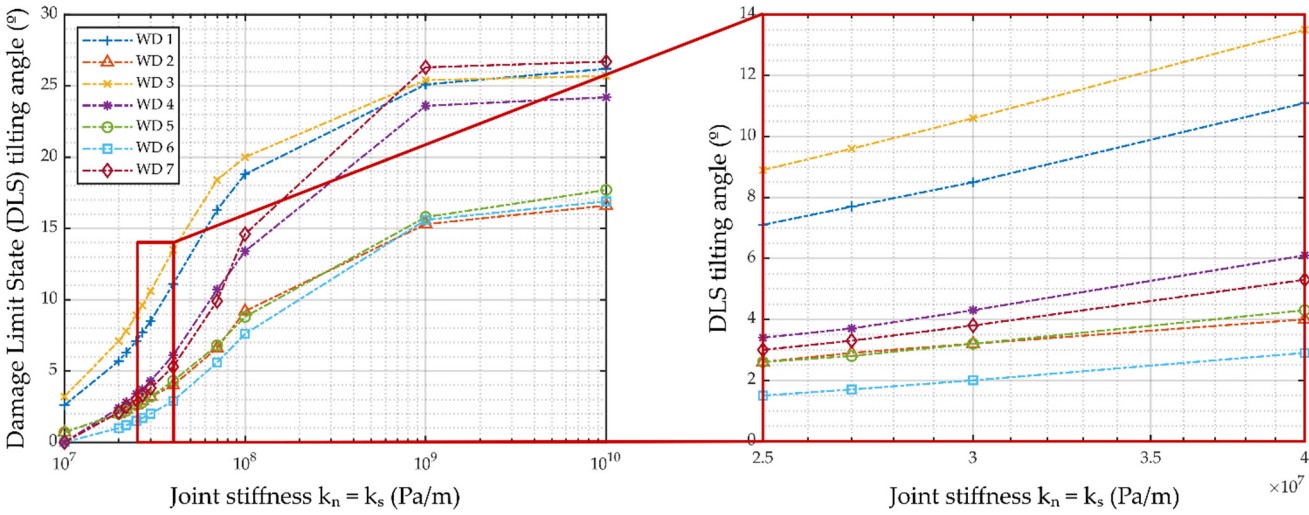

**Figure 12.** Evolution of the Damage Limit State (DLS) tilting angle for each wall design (WD) for different joint stiffness value. The drift limit considered here is 1%, i.e., the maximum one from the Eurocode 8 [36].

In Figure 12, it is noted that the DLS tilting angle is very small for low contact stiffness. In fact, all WDs follow a similar hyperbolic tangent trend, with three main different groups. WD 1 and WD 3 form the first one with the highest global stiffness in the low stiffness range resulting in smaller drifts. This is attributed to the even distribution of voids inside the structure, thus limiting stress concentration and joint rotation/deformation. Then, the second group (WD 4 and WD 7) and the third group (WD 2, WD 5 and WD 6) have the same behaviour at low contact stiffness (up to $k_n = k_s = 4 \times 10^7$ Pa/m), while the DLS collapse tilting angles of the second group increase at medium contact stiffness to reach similar values as the first group (WD 1 and WD 3) for joint stiffness of $k_n = k_s = 1 \times 10^9$ Pa/m. The third group (WD 2, WD 5 and WD 6) always has the smaller DLS tilting angles.

Table 4 compares the ratio of (1) the collapse tilting angle and (2) the DLS tilting angle for the simulation using $k_n = k_s = 3 \times 10^7$ Pa/m to the simulation using $k_n = k_s = 1 \times 10^9$ Pa/m for the seven WDs. It is shown that using an inaccurate value for the contact stiffness (in this case a larger one) overestimates not only the collapse tilting angle (by an average factor of 1.7, as seen before), but even more the DLS tilting angle ratio, by an average factor of 5.0. This last comment is of particular importance when assessing the seismic capacity of such structures.

**Table 4.** Comparison of the collapse and the damage limit state (DLS) tilting angles between simulations with $k_n = k_s = 3 \times 10^7$ Pa/m and $k_n = k_s = 1 \times 10^9$ Pa/m for each WD. The results are displayed in the form of the ratio between the stiffer to the softer joint stiffness.

|  | WD 1 | WD 2 | WD 3 | WD 4 | WD 5 | WD 6 | WD 7 |
|---|---|---|---|---|---|---|---|
| Collapse tilting angle ratio | 1.3 | 1.9 | 1.3 | 1.4 | 2.0 | 2.7 | 2.0 |
| DLS tilting angle ratio | 3.0 | 4.8 | 2.4 | 5.5 | 4.9 | 7.8 | 6.9 |

Although it has been excluded to simplify the parametric analysis, it is stressed that the accurate evaluation of the stiffness parameters concerns both normal $k_n$ and tangential

$k_s$ stiffness. However, as already noticed above, the effect of $k_s$ is relatively small for the studied configurations. As an illustrative example, Figure 13 shows the pushover curves of WD 6 and the DLS associated with a drift limit of 1% for different combinations of $k_n$ and $k_s$. Except for very small $k_s$ associated with large $k_n$, the influence of $k_s$ is again much less important than $k_n$, although always noticeable. This is attributed to the lower tangential contact forces compared to the normal ones as the tilting angles are almost always smaller than 20° for every WD ($\tan 20° = 0.36$).

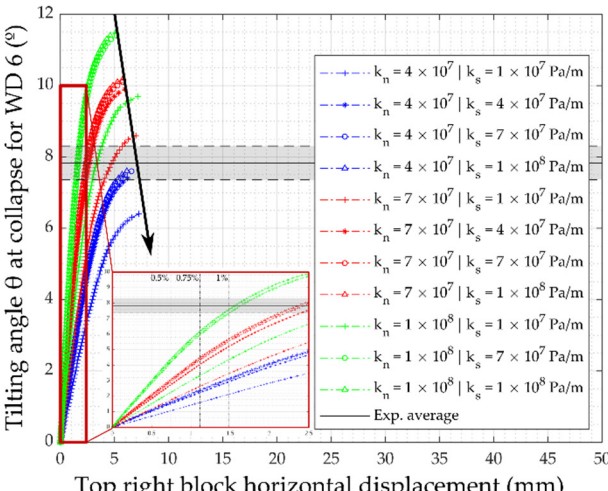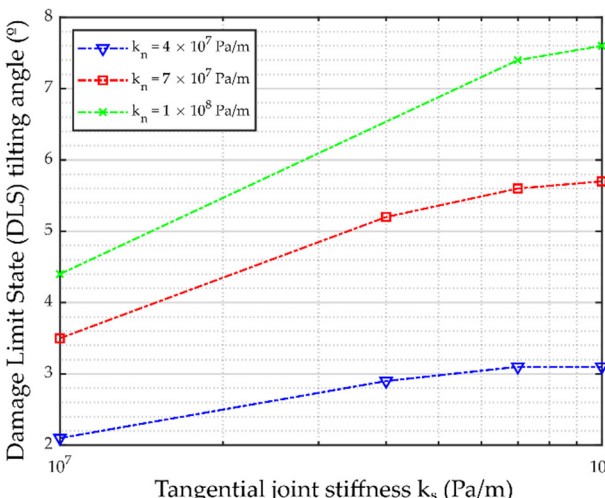

**Figure 13.** Influence of the tangential joint stiffness $k_s$ on the pushover curves and Damage Limit States (DLS) of WD 6 for different normal joint stiffness.

Finally, one can argue that the drift limit of 1% for this particular type of soft wall should be revised to account for most of the elastic part of the pushover curves (Figures 9–11). Actually, this drift limit should even depend on the joint stiffness of the assemblage (Figure 12). However, since the other drift limits of 0.5% and 0.75% from the Eurocode 8 are linked to the protection of the non-structural elements bound to the structural shear wall, they should be kept [36]. In addition, it also means that non-structural elements should be avoided on these kinds of very soft walls, as they will break too easily for very small and common earthquakes/loads.

## 6. Conclusions

This paper presented numerical DEM simulations of seven perforated masonry shear walls taken from an experimental work in the literature [22]. The work stemmed from the inaccurate predictions provided by Limit Analysis [27]. After the description and calibration of numerical parameters (contact points and density), a parametric analysis was conducted on the stiffness parameters. It was found that the tangential stiffness has only a minor effect on the collapse load and is limited to the range of very low stiffness ($1 \times 10^7$–$2 \times 10^7$ Pa/m). Classical stiffness values ($1 \times 10^9$–$1 \times 10^{10}$ Pa/m) achieved very similar results to Limit Analysis (either with associative or non-associative flow rule). In this case, the experimental capacity is overestimated by an average of 70%. Using a much smaller value of stiffness, which is classical in rubble masonry assemblages subjected to low vertical stresses and scaled models in the laboratory, DEM predictions were much closer to the results. A value of $k_n = k_s = 3 \times 10^7$ Pa/m was enabled to be, on average, only less than 10% different from the experimental failure value. The approach adopted seems more adequate than the one proposed by [27], where different geometrical parameters should be used for each geometrical model to fit the experimental capacity.

In general, simulations with classical ($1 \times 10^9$ Pa/m) and low stiffness ($3 \times 10^7$ Pa/m) values retrieved the experimental collapse mechanisms of the tested walls, though only the

simulations with low stiffness could reproduce the progressive bending of masonry piers due to the accumulated joint deformation at each masonry course.

In real structures, contact stiffness values are available in standards and from testing large scale structures/structural components. The results, particularly when performance based seismic assessment is adopted, are sensitive to the adopted stiffness. Stiffness values must be significantly decreased to reach accurate simulations when a structure is made of dry joints with imperfect blocks, as for the studied dry masonry walls or generally for dry-stone constructions, and particularly in the case of low vertical precompression. In this case, the stiffness parameters must be identified on the model or at least with joint closure tests, since it significantly affects the final capacity and ductility of the system. The error made using too stiff joints ($1 \times 10^9$ instead of $3 \times 10^7$ Pa/m) reaches a factor of 1.7 for the collapse load and a factor of 5.0 for the damage limit state (DLS) load based on the acceptable drift of 1% from the Eurocode 8. The present work also highlighted the inadequacy of the drift limits proposed by the Eurocode for this kind of soft assemblage. The latter requires a definition based on the stiffness of the structure, at least for the third category where no non-structural components are linked to the studied structural element.

Finally, though standard Limit Analysis (either with associative or non-associative flow rules) seems valid for masonry structures with classical joint stiffness ($1 \times 10^9$ Pa/m), it is not applicable as-is for softer joints with lower stiffness values, in which elastic deformation plays a key role in the response.

**Supplementary Materials:** The following are available online at https://www.mdpi.com/article/10.3390/app12042108/s1, Video S1: WD 1, Video S2: WD 2, Video S3: WD 3, Video S4: WD 4, Video S5: WD 5, Video S6: WD 6, Video S7: WD 7. They all represent the numerical simulations (with the displacement maps) for the joint stiffness of $k_n = k_s = 3 \times 10^7$ Pa/m.

**Author Contributions:** Conceptualisation, N.S.; methodology, N.S. and P.B.L.; investigation, N.S. and G.M.; resources, P.B.L. and G.M.; writing—original draft preparation, N.S.; writing—review and editing, P.B.L. and G.M.; visualisation, N.S.; funding acquisition, P.B.L. All authors have read and agreed to the published version of the manuscript.

**Funding:** This work was partly financed by FCT/MCTES through national funds (PIDDAC) under the R&D Unit Institute for Sustainability and Innovation in Structural Engineering (ISISE), under reference UIDB/04029/2020. This study has also been funded by the STAND4HERITAGE project (New Standards for Seismic Assessment of Built Cultural Heritage) that has received funding from the European Research Council (ERC) under the European Union's Horizon 2020 research and innovation program (Grant No. 833123) as an Advanced Grant. Its support is gratefully acknowledged. However, the opinions and conclusions presented in this paper are those of the authors and do not necessarily reflect the views of the sponsoring organisations.

**Institutional Review Board Statement:** Not applicable.

**Informed Consent Statement:** Not applicable.

**Data Availability Statement:** The geometrical models used to generate the numerical 3DEC models and the FISH and Python routines developed for the present study are publicly available on the associated Zenodo record [40].

**Acknowledgments:** The authors are grateful to A. Mehrotra who provided support to evidence the potential of DEM for these cases of study. The authors also acknowledge all STAND4HERITAGE fellows for the fruitful discussions related to this study.

**Conflicts of Interest:** The authors declare no conflict of interest. The funders had no role in the design of the study; in the collection, analyses, or interpretation of data; in the writing of the manuscript, or in the decision to publish the results.

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
