# Peer review of "Joint Stiffness Influence on the First-Order Seismic Capacity of Dry-Joint Masonry Structures: Numerical DEM Investigations"

_applsci, doi:10.3390/app12042108_

Round 1
Reviewer 1 Report
The paper presented numerical DEM simulations of seven perforated masonry shear walls taken from an experimental work of the literature and is stemmed from the inaccurate predictions provided by Limit Analysis. It is focused on the appropriate selection of joint stiffness to predict the maximum collapse tilting angle of each type of wall. At the final, this angle is compared to the drift ratio of the damage limit states of EN1998. It is well written and presents clear all the steps of the analyses. My advice is to accept the paper.
A suggestion to the authors is to write the power of e as a superscript. I also noticed that a link in line 189 is missing.
Reviewer 2 Report
The paper concerns the modelling of dry-joint masonry structures under lateral loads through DEM analysis. 7 scaled shear walls are modelled and the results are compored with both previous experimental tests (tilting tests) and previous numerical analysis based on Limit analysis. The sentitivity of the results from the stiffness of the dry joint is investigated: it is found that a reduced joint stiffness (attributable to masonry imperfections) better fits the experimental results in terms of collapse angle and failure mode.
The paper is well written and organized, contents are clearly exposed and understandable. The current state of the art is adequately referenced. The main assumptions are argumented and the results clearly discussed. The provided supplementary material (animations) is useful to improve the comprehension of the collapse phenomena.
Therefore, the paper is worth of publication after minor revision, in light of the following reviewer's comments:
SECTION 2: provide the dimensions of the bricks and wall thickness
SECTION 3:
- a figure with schematization of blocks and equivalent springs can help the readear in the comprehension of the model (with straggered bricks)
- only bed joints are modelled or also head joints? Please specify
- line 160: acceleration, mass (need a comma)
- equation 3: define symbol µ (friction coefficient)
- line 172: maybe, add a reference for classical local damping = 0.8
- in general, in all equations, consider to replace the multiplyer symbol "x" with "·" to increase readability
- figure 2: there is a error in the caption, please correct.
SECTION 4.2
- figures 4 and 5: please provide indication about the colours gradient
SECTION 4.3
It is necessary a clearer explaination for the assumtion ks=kn, since it is an important and critical aspect of the study. It seems that the tangential stiffness is not important (unexpectedly, from my side)... is it related to the low axial stress level? Does ks influence at least DLS? Please discuss more in deep in the text. Which combinations of ks and kn were tested? It can be useful to add some results of the sentisitivity analysis performed, to proved what claimed
- line 307: results for kn > 1e10 are not reported in Figure 6 (left side, not right). So, maybe it is not "evident" as written but it can be deduced from the curves trend.
- in caption of Table 3: please specify that the numerical values are for kn=ks=3e7 Pa/m
- in captions of Figure 7 and 8: 3e7 Pa/m
SECTION 5:
- although in the lack of experimental pushover curves, can the authors provide at least an indication of the experimental displacement at collapse? Or some other (even rougly) experimental information concerning ultimate stable displacement? This can considerably increase the significance and robustness of thier findings
- line 396: "on the left"?
- lines 439-443: maybe "unrealistic" instead of "wrong", since a precise, direct comparison with the experiments in terms of displacement is lack
REFERENCES
- ref. [21] incomplete
General observation:
- what is intended as masonry "imperfection" and "construction defect" deserves some better explaination in the manuscript, being a key aspect
- as the angle incrases, the axial stress level reduces, due to inclination: can this variation influence somehow the joint stiffness? Please comment in the manuscript
